# Fine Litter Dynamics in Tropical Dry Forests Located in Two Contrasting Landscapes of the Colombian Caribbean

Jeiner Castellanos-Barliza [1,2,*], Victoria Carmona-Escobar [1], Jean Linero-Cueto [3], Eber Ropain-Hernández [1] and Juan Diego León-Peláez [2]

[1] Grupo de Investigación en Restauración Ecosistémica y Ecología Urbana, Facultad de Ciencias Básicas, Universidad del Magdalena, Santa Marta D.T.C.H., Santa Marta 470002, Colombia; victoriacarmonave@unimagdalena.edu.co (V.C.-E.); eropainh@unimagdalena.edu.co (E.R.-H.)

[2] Grupo de Investigación en Restauración Ecológica de Tierras Degradadas en el Trópico, Universidad Nacional de Colombia, Calle 59A 63-20, Medellín 050034, Colombia; jdleon@unal.edu.co

[3] Facultad de Ingeniería, Universidad del Magdalena, Santa Marta D.T.C.H., Santa Marta 470002, Colombia; jlineroc@unimagdalena.edu.co

\* Correspondence: jcastellanos@unimagdalena.edu.co

**Abstract:** Tropical dry forests (TDFs) represent 42% of all tropical forests; they are extensive, but little is known of their structure and function. The fine litterfall represents the main route of circulation of organic materials and nutrients in these ecosystems. The objective of this study was to compare several remnants of TDFs located in contrasting landscape units—Mountain and Lomerio—and with different precipitation, in terms of the fluxes of organic materials to the soil, derived from the production of fine litterfall from the canopy. The fine litterfall (including woody material up to 2 cm in diameter) was collected monthly from April 2020 to March 2021, in 29 circular plots of 500 m$^2$ randomly established. High rates of litterfall were recorded in the Lomerio landscape (4.9 Mg ha$^{-1}$) than in the Mountain landscape (4.5 Mg ha$^{-1}$). The monthly leaf litter production showed clear seasonal patterns, which were largely driven by the importance of the species in the landscape and the effect of precipitation during the study. Annual fine litter production observed in this study in comparison with other TDFs indicates relevant productivity levels, which contribute to the activation of biogeochemical cycles and improved ecosystem functionality.

**Keywords:** biogeochemical cycles; mountain landscape; lomerio landscape; *Astronium graveolens*; *Pterocarpus acapulcensis*

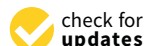



## 1. Introduction

Tropical forests play an important role in the dynamics of the global carbon cycle, particularly through their fixation from the atmosphere and subsequent accumulation in plant biomass and soil [1,2]. In fact, as a result of their interactions with the planetary climate system, these ecosystems support 30–40% of the net primary productivity of terrestrial ecosystems [3,4].

Among tropical forests, seasonally dry forests (TDFs) account for 42% of the world's tropical forests [5], are extensive, but little known in their structure and function compared to tropical rainforests [6,7]. In the Neotropical region, TDFs have been globally reduced by 49% to 66% of their original coverage, occurring in patches, immersed in landscapes dominated by crops and livestock areas [8,9].

Among the main nutrient cycling routes in tropical dry forests, the following stand out as nutrient inputs: biological nitrogen fixation, and atmospheric deposition by precipitation or dry deposition, which include gases and particulate matter, such as those from the Sahara [10,11]. However, the main route of nutrient cycling in the TDFs is represented by the fine litterfall from the canopy, which, after its decomposition, favors the gradual incorporation of organic materials of different nature into the soil [12,13].

Therefore, it supports net primary productivity, carbon stock accumulation, and soil fertility [14]. In addition, it represents one of the major indicators in the assessment of the recovery of ecosystem functions in degraded forests and can provide a basis for understanding the responses of forest ecosystems to climate change [15,16].

In particular, the potential returns of organic matter and nutrients to the soil through fine litterfall in TDFs differ quantitatively and qualitatively on a seasonal and annual basis for each site studied [17]. Furthermore, they can be affected by multiple factors such as soil nutrient availability, precipitation, the successional state of the vegetation, its structure, and composition [18–20]. In addition, return rates can be directly regulated by the production and chemistry of leaf litter and its foliar, reproductive, and woody components [15,21]. Therefore, monitoring of fine litter can be considered a key parameter in the study and recovery of ecosystem functions within TDFs.

The remnants of TDFs in the Colombian Caribbean cover an area of about 420,000 ha (data from [22]) and are highly degraded and transformed as a consequence of their replacement by productive systems and rapid urban growth, determining small, isolated patches as the dominant matrix of the landscape [23]. Consequently, the representation of tropical dry forests in this region is dominated by isolated patches of secondary forests and not by native forests, whose current conservation status is due to voluntary decisions by the owners of the areas where these patches are located.

This study was conducted in several remnants of these secondary forests in the department of Magdalena, where the current representation of this ecosystem is close to 23% (ca. 94,600 ha) of the total of the Colombian Caribbean. The objective of the study was to compare the fluxes of organic materials to the soil through fine litterfall in tropical dry secondary forests located in contrasting landscape units (Lomerio and Montaña). The Mountain landscape, unlike the Lomerio landscape, is characterized by steep slopes and higher rainfall, factors that potentially affect surface geomorphological processes and, consequently, the biogeochemical cycle and the forest soils located in them. In particular, given that the two types of landscape where these forests are located have a differential influence on the factors associated with the contribution of organic matter via fine litterfall, the following hypothesis was proposed: In the forests located in the Lomerio landscape, the highest fertility and soil clay content, resulting from less soil erosion compared to the Mountain landscape, favors higher rates of fine litter production. Although the amount of rainfall in both landscapes is different, it is expected that the main determinants of litter production rates result from the differentiation established by the type of landscape on the soil.

We hope that the results of this study will be useful for the design of management measures for the secondary forests of the Colombian Caribbean, thus contributing to the assurance of key ecosystem functions, particularly those associated with biogeochemical cycles.

## 2. Materials and Methods

### 2.1. Study Area

Sampling sites were located in two large landscape units, Mountain and Lomerio, in Magdalena, Colombia (Figures 1 and 2). In each landscape, two patches of secondary dry forest were selected (Mountain landscape: ARA 1 and ARA 2, and Lomerio landscape: PLA 3 and SBP 4). The landscapes presented clear differences in topography and precipitation regime. In particular, the Mountain landscape is characterized by the presence of high hills and steep slopes between 45–55%, a mean annual temperature of 27.4 °C and mean annual precipitation of 1551 mm, with a rainy period of eight months of higher precipitation from April to November (Figure 1). These characteristics favor the potential development of erosion processes of the land surface, as the surface runoff has a greater capacity to drag organic matter and soil particles towards the low areas. [24]. In contrast, the Lomerio landscape is characterized by low hills of lower slopes, between 5–10%, and moderate undulations. The mean temperature is 28.1 °C and the mean annual precipitation is 1112 mm, with a less intense rainy period of seven months (Figure 1, from April to

October). Under these conditions of topography and precipitation, the redistribution of rainwater potentially determines preferential flow, infiltration, and not surface runoff [25]. Consequently, in this landscape, unlike what happens in the Mountain landscape, potential soil erosion is lower, with accumulation processes predominating on the surface of the land. As a result of the influence of the different characteristics described for both landscapes, as well as the effect of the vegetation, the soils present differences in their physical and chemical properties (Table 1). In the Mountain landscape, the soils have sandy textures, slightly acidic pH, higher Bulk density, low P content, and high exchangeable bases, particularly high Calcium and Magnesium values. Consequently, these soils exhibit a high cation exchange capacity (CEC). In contrast, the soils of the Lomerio landscape have a clayey texture, moderately to slightly acidic pH, high contents of P, low exchangeable bases, and consequently a low cation exchange capacity (CEC).

In general, structural characteristics and species composition were similar in both landscapes. The vegetation is typical of the TDFs, dominated by trees that can reach between 10–12 m in height and closed canopy (Table 1), with the dominance in a greater proportion of species of the family Fabaceae (Leguminous plants), such as *Pterocarpus acapulcensis* Rose, *Machaerium goudotii* Benth. and *Acacia collinsii* Saff. In a lesser proportion appear the families Anacardiaceae, Capparaceae, Boraginaceae, and Malvaceae, such as *Astronium graveolens* Jacq., *Spondias mombin* L., *Quadrella odoratissima* (Jacq.) Hutch., Oken, *Cordia alliodora* Ruiz and Pav., *Guazuma ulmifolia* Lam. and *Pseudobombax septenatum* (Jacq.) Dugand.

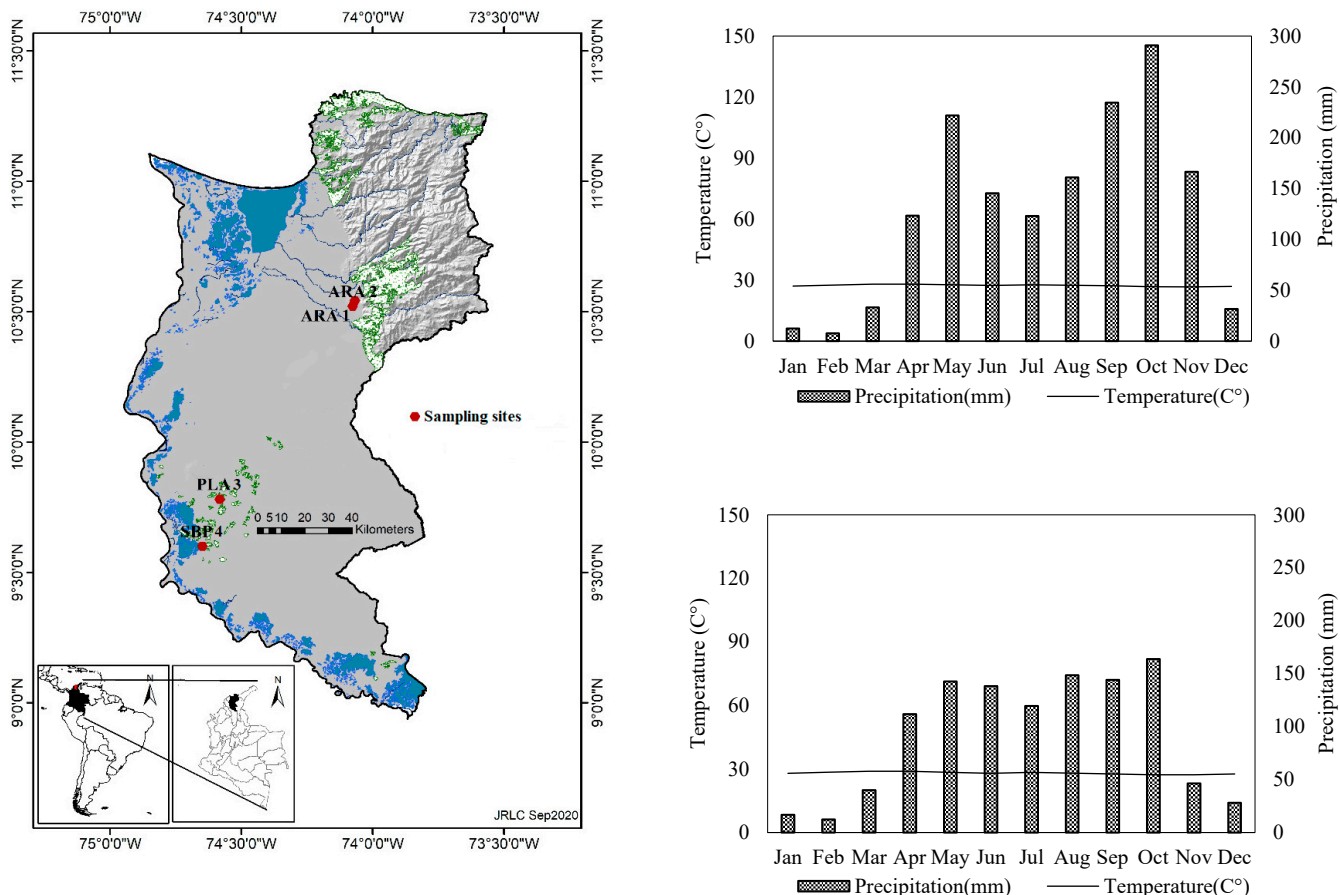

**Figure 1.** Study area and climatographs indicating air temperature and precipitation. Mountain landscape: ARA 1 and ARA 2; Lomerio landscape: PLA 4 and SBP 5. Annual precipitation and temperature were 1551 mm and 27.4 °C for the Mountain landscape, respectively, and 1112 mm and 28.1 °C for the Lomerio landscape, during the long-term record (1958–2019).

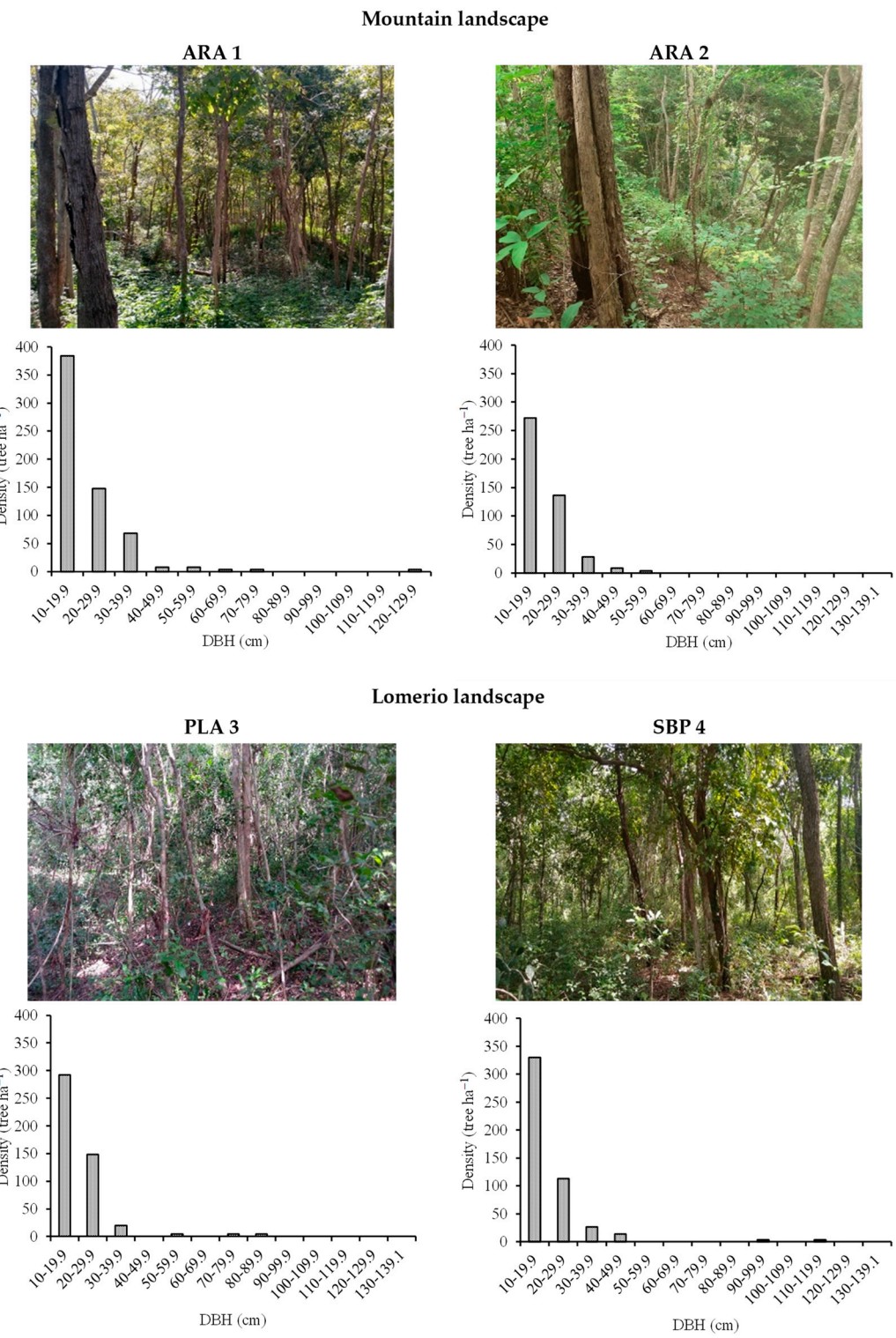

**Figure 2.** Landscape characteristics and diameter structure in TDFs in Magdalena, Colombia.

The dominant matrix around each TDFs presented high intervention by livestock activities and the presence of corn, bean, and cassava crops.

**Table 1.** Structural characteristics and physicochemical properties of soils (0–20 cm) in TDFs in Magdalena, Colombia. Mean $\pm$ SE ($n$ = 9 plots for Mountain landscapes, and $n$ = 12 plots for Lomerio landscapes). D: density (trees ha$^{-1}$) of trees with DBH (tree diameter at breast height) > 10 cm, G: basal area (m$^2$ ha$^{-1}$), H: height (m), IVI: importance value -index, CEC: cation exchange capacity, $\rho$: bulk density.

| Parameters | Structural Characteristics | | | |
| | Mountain Landscape | | Lomerio Landscape | |
| | ARA 1 | ARA 2 | PLA 3 | SBP 4 |
|---|---|---|---|---|
| Age (years) | >40 | >40 | >40 | >40 |
| D (tree ha$^{-1}$) | 628 $\pm$ 49.80 | 444 $\pm$ 62.21 | 468 $\pm$ 40.99 | 490 $\pm$ 50.83 |
| DBH mean (cm) | 21 $\pm$ 1.33 | 19 $\pm$ 1.24 | 19 $\pm$ 1.21 | 20 $\pm$ 2.54 |
| G (m$^2$ ha$^{-1}$) | 26 $\pm$ 3.92 | 14 $\pm$ 2.10 | 17 $\pm$ 2.97 | 20 $\pm$ 4.32 |
| H (m) | 11 $\pm$ 0.65 | 10 $\pm$ 0.74 | 12 $\pm$ 1.27 | 11 $\pm$ 0.63 |
| | *A. graveolens* (27) | *P. acapulcensis* (50) | *C. baducca* (68) | *P. acapulcensis* (55) |
| | *R. chryseum* (23) | *M. goudotii* (29) | *Q. odoratissima* (60) | *A. graveolens* (53) |
| Dominant species (IVI) | *A. inermis* (22) | *A. graveolens* (17) | *A. graveolens* (41) | *S. mombin* (32) |
| | *C. alliodora* (19) | *S. mombin* (16) | *P. septenatum* (17) | *A. collinsii* (24) |
| | *T. oblonga* (18) | *G. ulmifolia* (16) | *C. alliodora* (13) | *P. pinnatum* (16) |
| | Soil physicochemical characteristics | | | |
| BD (g/cm$^3$) | 1.5 $\pm$ 0.05 | 1.3 $\pm$ 0.10 | 1.1 $\pm$ 0.08 | 1.1 $\pm$ 0.03 |
| Sand (%) | 65.6 $\pm$ 1.60 | 64.8 $\pm$ 4.07 | 18.0 $\pm$ 2.97 | 3.9 $\pm$ 1.27 |
| Silt (%) | 18.4 $\pm$ 0.74 | 16.7 $\pm$ 0.63 | 27.7 $\pm$ 1.31 | 22.0 $\pm$ 2.40 |
| Clay (%) | 16.1 $\pm$ 1.06 | 18.5 $\pm$ 3.10 | 54.3 $\pm$ 4.08 | 74.1 $\pm$ 3.14 |
| pH (1:2) | 6.4 $\pm$ 0.50 | 6.5 $\pm$ 0.24 | 6.4 $\pm$ 0.23 | 5.7 $\pm$ 0.10 |
| C (%) | 1.6 $\pm$ 0.23 | 1.3 $\pm$ 0.15 | 1.9 $\pm$ 0.33 | 1.6 $\pm$ 0.23 |
| N (%) | 0.2 $\pm$ 0.02 | 0.2 $\pm$ 0.02 | 0.2 $\pm$ 0.04 | 0.2 $\pm$ 0.17 |
| C/N | 9.4 $\pm$ 11.0 | 8.5 $\pm$ 9.7 | 8.1 $\pm$ 7.6 | 9.6 $\pm$ 1.4 |
| P (mg/kg) | 4.9 $\pm$ 0.44 | 5.7 $\pm$ 2.42 | 24.1 $\pm$ 6.30 | 35.3 $\pm$ 4.14 |
| N/P | 400.5 $\pm$ 1.23 | 293.5 $\pm$ 0.18 | 116.7 $\pm$ 0.40 | 56.0 $\pm$ 0.12 |
| K (cmol (+)/kg) | 0.2 $\pm$ 0.05 | 0.2 $\pm$ 0.10 | 0.7 $\pm$ 0.14 | 0.8 $\pm$ 0.23 |
| Ca (cmol (+)/kg) | 3.2 $\pm$ 1.00 | 2.6 $\pm$ 0.84 | 12.0 $\pm$ 2.50 | 7.8 $\pm$ 0.91 |
| Mg (cmol (+)/kg) | 0.8 $\pm$ 0.07 | 0.9 $\pm$ 0.40 | 7.0 $\pm$ 0.60 | 7.1 $\pm$ 0.70 |
| CEC (cmol (+)/kg) | 4.7 $\pm$ 0.77 | 3.8 $\pm$ 0.86 | 19.8 $\pm$ 2.11 | 16.2 $\pm$ 1.46 |

*2.2. Sampling Design and Monitoring*

A total of 21 circular plots of 500 m$^2$ were randomly established in TDFs. Nine plots were distributed in the Mountain landscape (4 plots in ARA 1, and 5 plots in ARA 2), and 12 in the Lomerio landscape (6 plots in PLA 3, and 6 plots in SBP 4). Within each plot, all tree individuals $\geq$10 cm DBH (diameter at breast height: 1.3 m) were identified and measured. Species that could not be identified in the field had leaf, flower, or fruit samples collected, photographed, and subsequently identified in the herbarium of the University of Magdalena. The height (H) of each individual was recorded and some characteristics of the terrain and intervention activities were described.

For a period of 12 months (April 2020–March 2021), the production of fine litterfall was monitored monthly. A total of 150 litterfall traps were installed, distributed as follows: (35) ARA 1, (40) ARA 2, (40) PLA 3, and (35) SBP 4. The traps consisted of circular hoops of 0.5 m$^2$ and were constructed with fine mesh fabric [8]. The collected material was taken to the laboratory, separated into different fractions: Total leaf litter, Reproductive material (flowers and fruits, and their constituent parts), Woody material (up to <2 cm in diameter), and Other remains (unidentifiable or miscellaneous material). The foliar contributions of the four most dominant tree species in the landscapes were also individually separated: Leaf litter *A. graveolens*, Leaf litter *P. pinnatum*, Leaf litter *C. alliodora*, Leaf litter *P. acapulcensis*, and the contributions of the leaf litter of the other species. Subsequently, the separated material was subjected to the oven at a temperature of 65 °C, until a constant dry weight was obtained. The dry weight of each fraction was recorded and the sum of the weights of all the individual fractions represented the total weight of the fine litterfall.

Composite soil samples were collected from three plots from each TDF, six in total for each landscape at a depth of 0–20 cm. These samples were packed, marked, and sent to the laboratory of the International Center for Tropical Agriculture in Palmira, for the determination of texture (clay, silt, and sand; Bouyoucos method), carbon, and nitrogen by combustion in Elemental Analyzer. Available soil P was determined by the Bray II/ L-ascorbic acid method and colorimetry; Bulk density (ρ) was determined by taking undisturbed soil samples in the field (0–20 cm depth), through the ring method [26]. Subsequently, they were taken to the laboratory and dried at 105 °C for 48 h. The BD was obtained by dividing the dry mass of the soil (Mss) by the volume of the cylinder (g cm$^{-3}$). Soil pH was determined by a glass electrode (soil: water ratio,1:2). The exchangeable $Ca^{2+}$, $Mg^{2+}$, $K^+$, and $Na^+$ were extracted with 1 M ammonium acetate solution at pH 7 and their concentrations were measured with flame atomic absorption spectroscopy. The Cation Exchange Capacity (CEC) was determined by the sum of $Ca^{2+}$ + $Mg^{2+}$ + $K^+$ + $Na^+$ + $Al^{3+}$ + $H^+$ [26].

### 2.3. Data Analyses and Processing

Student's *t*-tests were performed to determine significant differences ($p < 0.05$) between landscapes for each of the fine litter fractions. Spearman correlation coefficients were calculated to test the correlation between total precipitation and monthly production patterns of fine litterfall fractions. Principal Component Analysis (PCA) was performed to explore the relationships between fine litter production variables (LL: total leaf litter; FL: total fine litter), structural characteristics of the TDFs (G: Basal area, H: height, D: tree density) and environmental characteristics (Soil: pH, Clay, Sand, CEC, C, N; Climate: precipitation). All statistical analyses were performed using Statgraphics Centurion XVII software and R version 3.5.3 (Vegan package).

### 2.4. Climatic Data

Precipitation and temperature data were obtained from the historical record between 1958 and 2019 from [27] and from the CHIRPS global database. The latter manages a quasi-global dataset (covering the area between 50° N and 50° S) developed by the U.S. Geological Survey Earth Resources Observation and Science Center CHIRPS available online (http://chg.geog.ucsb.edu/data/chirps/; accessed on 7 December 2021). Details of the CHIRPS database can be found at [28].

## 3. Results

### 3.1. Annual Fine Litterfall

Significant differences were observed in the annual rate of fine litter production between landscapes (Table 2, $p \leq 0.05$). Particularly, higher rates were recorded in the Lomerio landscape (4.9 Mg ha$^{-1}$) than in the Mountain landscape (4.5 Mg ha$^{-1}$). Total fine litterfall was dominated by the leaf fraction, representing more than 70% for both landscapes. Among dominant species, higher leaf production rates were observed for *A. graveolens* (0.5 Mg ha$^{-1}$ y$^{-1}$) and *P. acapulcensis* (0.3 Mg ha$^{-1}$ y$^{-1}$) in the Lomerio landscape. In contrast, lower values were observed for *C. alliodora* (0.01 Mg ha$^{-1}$ y$^{-1}$) in Mountain and *P. pinnatum* (0.03 Mg ha$^{-1}$ y$^{-1}$) in Lomerio.

**Table 2.** Annual contributions of fine litterfall (kg ha$^{-1}$ y$^{-1}$) in landscapes of Magdalena, Colombia. The similar letters indicate no statistically significant differences ($p > 0.05$). Mean ± SE ($n$ = 9 plots for Mountain landscapes, and $n$ = 12 plots for Lomerio landscapes).

| Fine Litter Fractions | Landscapes | |
| --- | --- | --- |
| | Mountain | Lomerio |
| Leaf litter *A. graveolens* | 115.5 ± 38.97 b | 529.9 ± 180.97 a |
| Leaf litter *P. pinnatum* | 138.1 ± 43.69 a | 10.9 ± 3.48 b |
| Leaf litter *C. alliodora* | 31.1 ± 7.26 a | 83.5 ± 27.95 a |
| Leaf litter *P. acapulcensis* | 312.3 ± 85.08 a | 371.7 ± 157.87 a |
| Leaf litter other species | 2733.2 ± 116.97 a | 2698.4 ± 219.85 a |

**Table 2.** *Cont.*

| Fine Litter Fractions | Landscapes Mountain | Lomerio |
|---|---|---|
| Total leaf litter | 3330.2 ± 114.06 b | 3691.4 ± 130.57 a |
| Reproductive material | 164.1 ± 41.54 a | 71.1 ± 18.85 b |
| Woody material | 502.9 ± 31.61 a | 552.2 ± 28.87 a |
| Other remains | 533.4 ± 78.08 a | 574.9 ± 39.21 a |
| Total fine litterfall | 4530.6 ± 175.43 a | 4889.5 ± 160.17 b |

*3.2. Litterfall Seasonality*

Significant correlations were recorded ($p \leq 0.05$) between total precipitation and monthly production patterns of fine litter fractions (Figure 3). Specifically, negative correlations were observed for Other remains (r = −0.63; $p = 0.0376$) in the Mountain landscape, and leaf litter (r = −0.90; $p = 0.0028$) and reproductive material (r = −0.61; $p = 0.0436$) for the Lomerio landscape. In general, for both landscapes, all fractions registered higher monthly production peaks between December and March during the months of lower precipitation and lower between April and November during the months of higher precipitation (Figure 3).

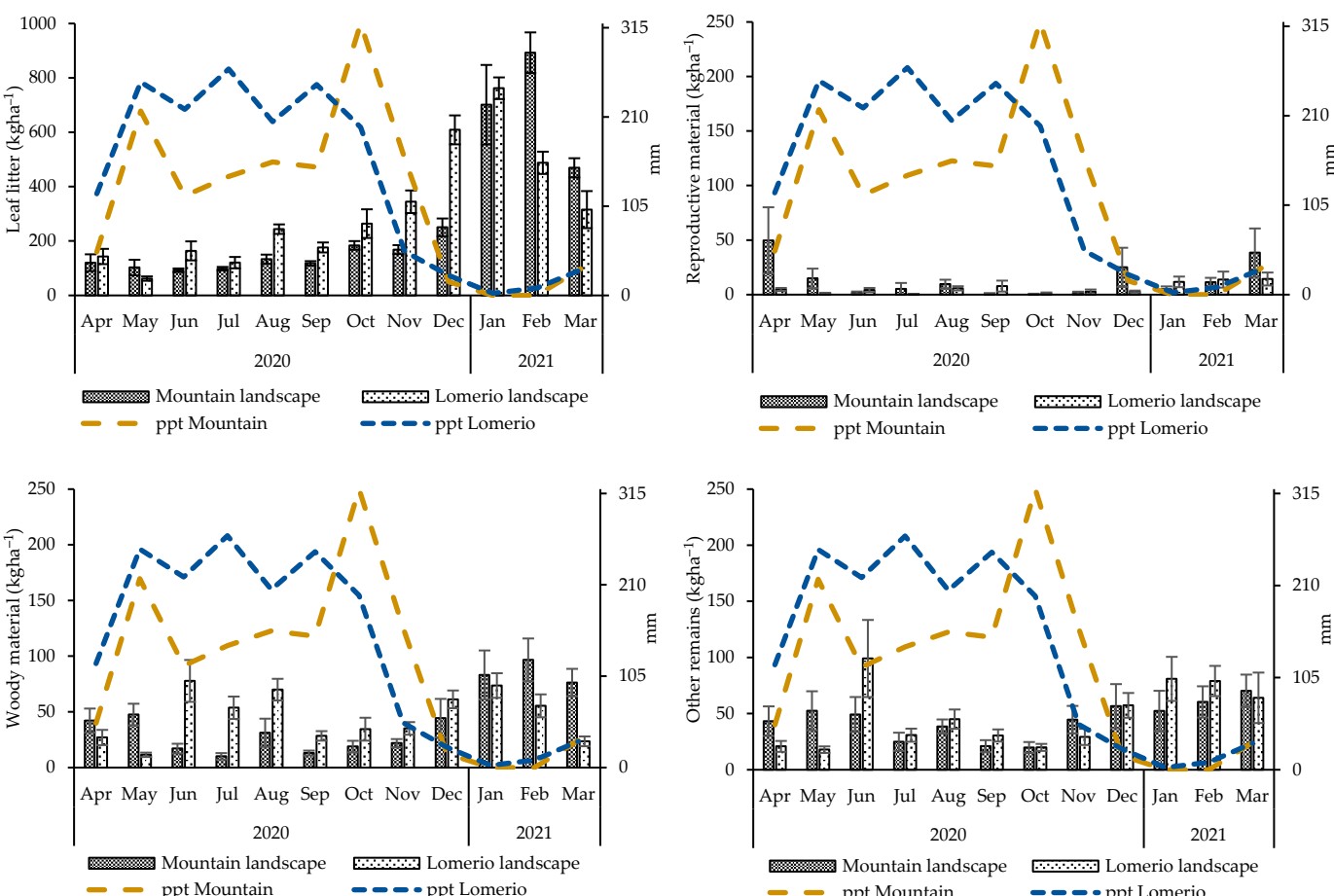

**Figure 3.** Monthly production of litter fractions (kg ha$^{-1}$) in landscapes of Magdalena, Colombia. ppt: precipitation. Mean ± SE (*n* = 9 plots for Mountain landscape, and *n* = 12 plots for Lomerio landscape).

Different patterns in the monthly distribution of leaf litter among dominant species were observed (Figure 4). Leaf litterfall of *A. graveolens* (r = −0.75; $p < 0.0125$), *P. acapulcensis* (r = −0.60; $p = 0.0483$) and *P. pinnatum* (r = −0.88; $p < 0.0035$) showed a negative corre-

lation with total precipitation in the Mountain landscape. Additionally, leaf litterfall of *A. graveolens* (r = −0.83; *p* < 0.0058) and *C. alliodora* (r = −0.66; *p* = 0.0276) for the Lomerio landscape. Thus, maximum values between October and March and lower values between April and September were observed for both species in the months of less precipitation. On the other hand, the species *P. pinnatum* and *C. alliodora* recorded lower and almost constant contributions throughout the year of study.

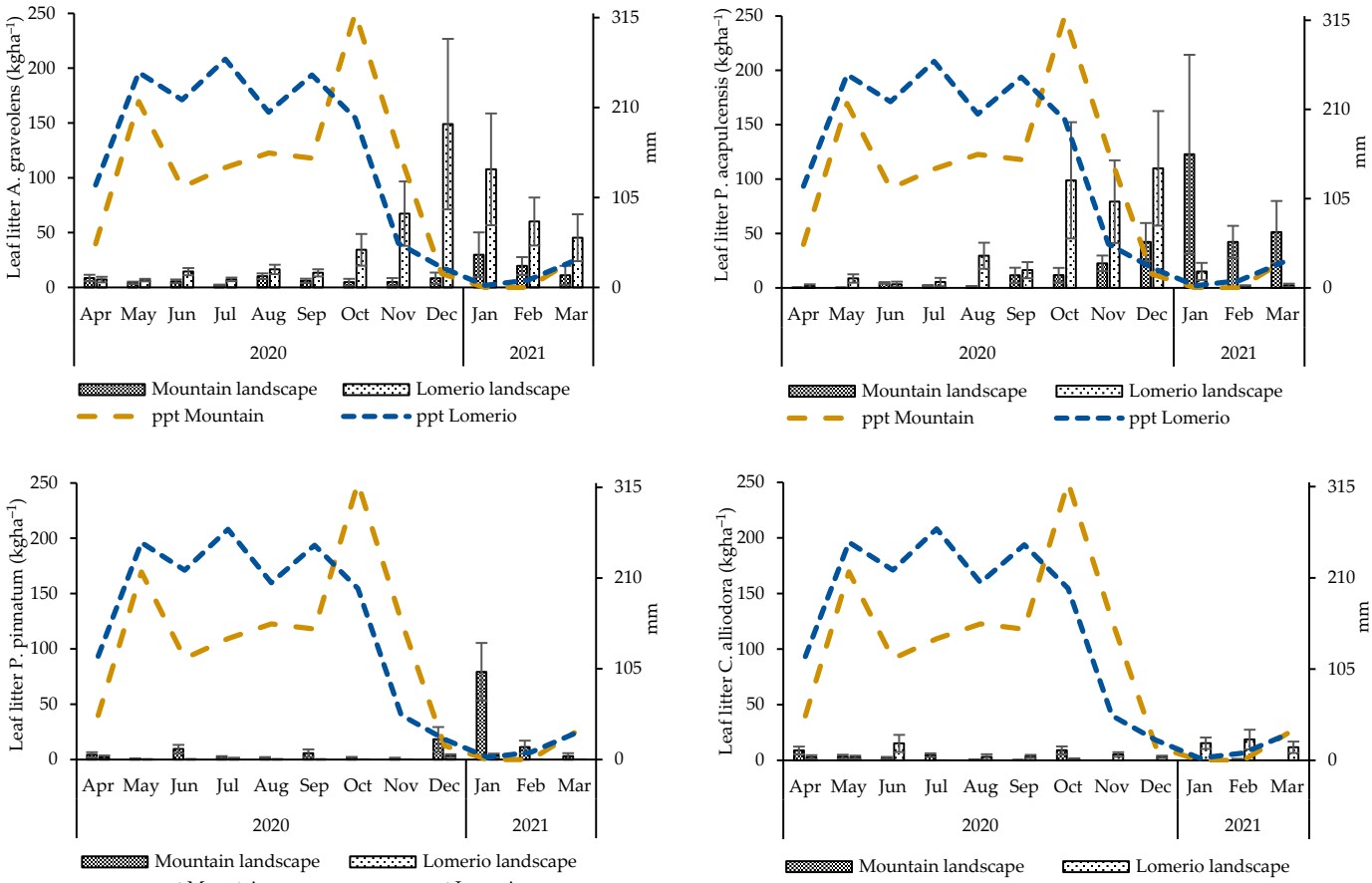

**Figure 4.** Monthly production of Leaf litter (kg ha$^{-1}$) of the dominant species in landscapes of Magdalena, Colombia. ppt: precipitation. Mean ± SE (*n* = 9 plots for Mountain landscapes, and *n* = 12 plots for Lomerio landscapes).

The first four components (PCA) explained 91.4% of the total variation (Figure 5). The 46.9% of the variability was explained by the first axis (PCA 1) and 20.6% by the second (PCA 2). In general, differences between the landscapes were observed along the ordination axes. For the first component (PCA 1), positive associations were observed for ppt (r = 0.44), Clay (r = 0.42), CEC (r = 0.41) and negative for Sand (r = −0.44). The second component (PCA 2) was positively associated with C (r = 0.57), N (r = 0.49), and negatively with pH (r = −0.32).

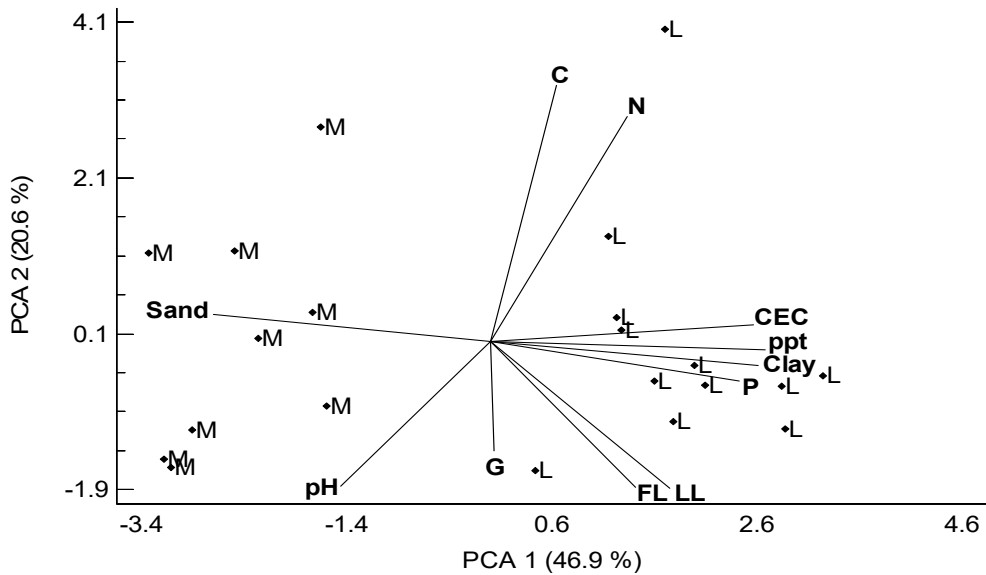

**Figure 5.** Correlation between the two components (PC 1 and PC 2) derived from two Principal Component Analyses of the landscapes in Magdalena, Colombia. M: Mountain landscape; L: Lomerio landscape; C: soil carbon content; P: Available soil P; ppt: precipitation; N: soil nitrogen content; CEC: cation exchange capacity; G: stand basal area; FL: total fine litterfall; LL: total leaf litter; ppt: precipitation.

## 4. Discussion

The present study showed the importance of monitoring fine litter production as a functional ecological process from the landscape point of view in secondary dry forests of the Colombian Caribbean. Information at this scale is scarce and local anthropogenic factors such as agriculture and cattle ranching can affect both short and long-term inputs of organic materials by this route. Consequently, this may have strong implications on nutrient cycling that supports net primary productivity, carbon stock, and soil fertility in TDFs [14,29].

The results of this study partially supported our hypothesis, which states that soil conditions in the landscape were determinant for the high litterfall rate. However, precipitation also had a strong effect on the litterfall inputs and their seasonal patterns during the study. Therefore, although different studies indicate that climate, soil fertility, structural characteristics, species composition, anthropogenic intervention, among others, influence the rate of fine litter production [14,19,30–32]; we consider the synergistic effect of these multiple factors to a greater or lesser extent over time. For this study, the combined effect of favorable soil conditions (fertility, particularly high soil P content and low soil N/P ratio) in the Lomerio landscape, and the higher precipitation registered there during the study period, marked the differential effect on fine litterfall rates and, particularly, on leaf litter inputs that fell to the soil.

### 4.1. Annual Fine Litterfall

The fine litter production rates obtained in this study represent high contributions of organic materials in both landscapes. In particular, the contributions from the foliar fraction (LL > 70% of FL), besides being the most abundant in this study, represent labile materials of high decomposability [33], which can result in higher inputs of organic materials and nutrients to the system. Therefore, such inputs result, among other aspects, in the improvement of the physicochemical properties of the first horizons of the soil [8,34], as was observed in the soil characteristics of the TDFs of the Lomerio landscape (Figure 2). On the other hand, the fine litter production rates determined in this study were higher than those recorded for other mature TDFs in Colombia. (0.3–3.5 Mg ha$^{-1}$ año$^{-1}$; [8,33],

and represented intermediate ranges for other dry and very dry tropical ecosystems in the world (1.2–8.7 Mg ha$^{-1}$ año$^{-1}$; [19,31,35–38]).

The WM, RM, and OR fractions represented minority contributions in both landscapes (2–12%). Similar patterns have been reported in different studies in a wide range of tropical forests [35,39–42]. However, the ecological importance of these fractions in the ecosystem cannot be underestimated, as they represent permanent sources of materials that are gradually incorporated into nutrient cycles. On the other hand, the permanent contribution of the RM fraction during this study suggests a seed bank potentially available in all TDFs, thus improving the regeneration capacity and diversity of seedlings present in these ecosystems [15].

*4.2. Litterfall Seasonality*

During this study, between 68% and 75% of foliar inputs were recorded in the dry season between November and March for the Lomerio and Mountain landscape, respectively. These patterns correspond to those recorded in other TDFs [19,43,44], where it is described that after the highest rainfall peaks (rainy season), leaf litterfall peaks follow. It is evident then that the greatest contributions of leaf litter coincided with the driest periods after the months of highest rainfall, in which temperature and evaporation increased, and during which the trees lost their leaves to avoid water loss [35,45].

In this study, the highest LL and FL inputs were associated with the best soil conditions (high clay and CEC contents) in the Lomerio landscape. These results can be explained by the feedback effect between litter production, litter decomposition, vegetation development, and soil fertility. Particularly in fertile soils, leaf turnover, leaf fall, and decomposition rates are rapid due to low energy investment for tissue synthesis and secondary metabolites [42]. Patterns similar to those obtained in our study have been observed in other TDFs [8,19].

Although historically, higher precipitation values were recorded in the Mountain landscape compared to the Lomerio landscape (Figure 1). Particularly, inverse patterns were observed during the study period (Figure 3). This is likely to have affected litter production rates at the landscape level, which was reflected in lower inputs of organic materials from the canopy to the soil in the Mountain landscape. This may have potential effects overtime on the return of organic materials and the biogeochemical cycling of soil nutrients, processes favored by the increased activity of microorganisms in litter decomposition and favored by the high humidity and temperature conditions [42].

Overall, leaf litter production of the species showed clear seasonal patterns, which were largely driven by the importance of the species in the landscape (IVI) and the effect of precipitation during the study (Figures 2 and 4). In this same sense, foliar inputs of *A. graveolens* represented between 2.5–14% of the total LL and between 9.4–10.1% for *P. acapulcensis*. These results reflect the large inputs of organic materials of these species from the canopy to the soil, which can potentially enter the system and provide a constant source of energy to soil microorganisms [33]. Hence, based on these attributes associated with productivity, both species can be projected as potential species for inclusion in future restoration programs. On the other hand, although the leaf litter production of *P. pinnatum* and *C. alliodora* species represented minority contributions of the total LL (0.3–3.0%), these fractions represent potential materials that permanently input the system.

## 5. Conclusions

Seasonal inputs and patterns of litterfall were strongly associated with precipitation and soil fertility conditions. These marked a differential response between the landscapes; around 10% of the leaf material contributed from the canopy in the Lomerio landscape. On the other hand, the high rate of fine litterfall observed in both landscapes in comparison with other TDFs indicates relevant productivity levels, which contribute to the activation of biogeochemical cycles and improve ecosystem functionality. In this sense, the results of our study were consistent in showing the importance of monitoring litter production as a functional indicator of the ecosystem and of the activities that promote the conservation

of these forests, which represent one of the few remaining TDFs found in the Magdalena region of Colombia, and which are immersed in a livestock matrix that threatens their permanence over time. On the other hand, of the four dominant species studied here, *A. graveolens* and *P. acapulcensis* stood out for their high rates of fine litter production. This aspect of their productivity allows them to be preliminarily identified as potential species for use in restoration and management programs for dry ecosystems degraded by human activities.

**Author Contributions:** Conceptualization, J.C.-B. and J.D.L.-P.; methodology, J.C.-B.; software, J.C.-B.; validation, J.D.L.-P. and J.C.-B.; formal analysis, E.R.-H.; investigation, V.C.-E.; resources, J.L.-C.; data curation, J.D.L.-P.; writing—original draft preparation, J.C.-B.; writing—review and editing, J.C.-B. and J.D.L.-P.; visualization, E.R.-H.; supervision, J.L.-C.; project administration, J.L.-C. funding acquisition, J.C.-B. All authors have read and agreed to the published version of the manuscript.

**Funding:** This research was funded by MINCIENCIAS: Fondo Nacional de Financiamiento para la Ciencia, la Tecnología y la Innovación "Francisco José de Caldas" (Postdoctoral Fellowship Program, 848 of 2019) and Universidad del Magdalena.

**Data Availability Statement:** Data are available from corresponding author upon reasonable request.

**Acknowledgments:** We thank the Sistema General de Regalías Fondo de Ciencia, Tecnología e Innovación del departamento del Magdalena, under the project "Investigación de los Efectos de la Variabilidad Climática y el Cambio Climático sobre el Recurso Hídrico, Biodiversidad y Actividades Agropecuarias en el departamento del Magdalena".

**Conflicts of Interest:** The authors declare no conflict of interest.

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
