# Peer review of "Fine Litter Dynamics in Tropical Dry Forests Located in Two Contrasting Landscapes of the Colombian Caribbean"

_forests, doi:10.3390/f13050660_

Round 1

Reviewer 1 Report

The study “Fine litter dynamics in tropical dry forests located in two contrasting landscapes of the Colombian Caribbean” is an interesting investigation on the composition of fine litter in two tropical dry forests with contrasting topography and rainfall. Also, this study needs to say more about the differences in topography and how this can affect soil nutrient conditions because the forest with more slopes would have more runoff. What about tree species with symbiotic bacteria? How different are these two forests in species composition? The contrast between forest is only for the topography and rainfall regimes?

Minor comments

L43. Change to ‘one of the main routes of nutrient circulation…’. Authors have not mentioned nitrogen fixation or atmospheric deposition (Saharan dust, for instance).

L54. Lawrence 2005 is the number 16; it is duplicated then.

LL64-70. This objective needs to have more information. Different precipitation says nothing, which one of the forests has higher precipitation. What does ‘in terms of their functioning’ mean?

LL70-72. The hypothesis needs more than just the soil characteristics. Canopy cover is the same? Species composition? Topography? Water and nutrients runoff?

LL72-74. Again, precipitation is different… Authors need more substantial arguments to formulate this hypothesis.

Figure 1. The map needs more sharpness; the name of the sites can barely be seen. Also, for the climatographic, the temperature axis needs to be realistic; the range from 0 to 150 °C is not.

Table 1. Spell out the acronyms D, DBH, G, H, IVI, CEC, etc., at the end of the table of in the legend. Some of these are not necessary and many are only used once (UTM, H, BD, and so on).

L138. Jaramillo.

Figures 3 -4. Precipitation is the same for all panels, but they look different because of the different scales. Only one single figure for precipitation would be better. Although, why some bars look different?

Author Response

COMMENTS TO THE AUTHOR

AUTHOR´S RESPONSE IN MS. WITH TRACK CHANGES

Reviewer # 1

The study “Fine litter dynamics in tropical dry forests located in two contrasting landscapes of the Colombian Caribbean” is an interesting investigation on the composition of fine litter in two tropical dry forests with contrasting topography and rainfall. Also, this study needs to say more about the differences in topography and how this can affect soil nutrient conditions because the forest with more slopes would have more runoff. What about tree species with symbiotic bacteria? How different are these two forests in species composition? The contrast between forest is only for the topography and rainfall regimes?

We agree with the reviewer's suggestion. Particularly,

in the Study Area section, the differences between the two landscapes studied in terms of topography, precipitation regime, and physical-chemical properties of the soil were better described. Likewise, the description of the forests present in the two types of landscape was improved, in terms of their composition of species and dominant families. L86-118.

L43. Change to ‘one of the main routes of nutrient circulation…’. Authors have not mentioned nitrogen fixation or atmospheric deposition (Saharan dust, for instance).

The reviewer's suggestion was accepted. The following text was included: Among the main nutrient cycling routes in tropical dry forests, the following stand out as nutrient inputs: biological nitrogen fixation, and atmospheric deposition by precipitation or dry deposition, which include gases and particulate matter, such as those from the Sahara (Johnson & Turner, 2019; Van Langenhove et al. 2020). L43-46.

L54. Lawrence 2005 is the number 16; it is duplicated then.

The reviewer's suggestion was accepted, and the reference was deleted.

LL64-70. This objective needs to have more information. Different precipitation says nothing, which one of the forests has higher precipitation. What does ‘in terms of their functioning’ mean?

All changes were made as the reviewer suggested. The phrase 'in terms of their functioning' was eliminated from the text and the differences between the landscapes studied were explained more precisely and deeply. In the same sense, it is noted that these differences between the two landscapes are described in greater depth in the methodology. L67-79 and L86-118.

LL70-72. The hypothesis needs more than just the soil characteristics. Canopy cover is the same? Species composition? Topography? Water and nutrients runoff?

The reviewer's suggestions were accepted. Particularly, the hypothesis was modified, explaining how the dominant surface geomorphological processes within each landscape affect the soil and its relationship with the production of fine litter in them. In order to offer a better explanation of the aspects questioned in the Hypothesis by the reviewer, it is necessary to indicate that in the Study Area section (L86-118), different aspects are already clarified (ie topography, physical-chemical properties soil, precipitation regime, species composition and structural characteristics of the forests) related to the differences between the two landscapes.

LL72-74. Again, precipitation is different… Authors need more substantial arguments to formulate this hypothesis.

The reviewer's suggestion was accepted. It is noted that these arguments for testing the hypothesis were expanded and better described in the Introduction and Methodology section. L67-79 and L86-118.

Figure 1. The map needs more sharpness; the name of the sites can barely be seen. Also, for the climatographic, the temperature axis needs to be realistic; the range from 0 to 150 °C is not.

The reviewer's suggestions are accepted. Figure 1 was edited and improved. On the other hand, it is clarified that the value of the axis is the correct one, since in the construction of the climadograms the methodology suggests that visually the values of the temperature axis are half of the total values of the precipitation axis.

Table 1. Spell out the acronyms D, DBH, G, H, IVI, CEC, etc., at the end of the table of in the legend. Some of these are not necessary and many are only used once (UTM, H, BD, and so on).

The reviewer's suggestions were accepted and the changes were made. See Table 1.

L138. Jaramillo.

We agree with the reviewer's suggestion. We change the reference to the journal style. L173

Figures 3 -4. Precipitation is the same for all panels, but they look different because of the different scales. Only one single figure for precipitation would be better. Although, why some bars look different?

We agree with the reviewer's suggestion.

Figure 3 and 4 were edited to make the axes look similar. On the other hand, it should be noted that both figures present the same precipitation but different contributions in the fine litter and species fractions. For this reason, the bars are of different sizes. For better visualization and clarity of the figure, it is not possible to make a single figure containing all the litterfall fractions evaluated in this study.

Reviewer 2 Report

  1. Clarify what you mean by” Seasonally Dry Forests”?

  1. What do you mean by “Fine litterfall” Is there a such a thing as  “non Fine litterfall”?

  1. Materials and Methods. The mean temperatures and annual precipitation were derived from which period?

  1. Change the “Mountain landscape was characterized by the presence of high hills into “  Mountain landscape is characterized by the presence of high hills into  “ 

  1. Change the “The Lomerio landscape was characterized by the presence of high hills into “The Lomerio landscape is characterized by the presence of high hills into..”

  1. Change the “the soil properties..” into the soil properties of these sites are slightly acidic, have a sandy texture, have a low P content and a low cation….”

  1. The concentrations of base cations in Table 1 are written as : cmol (+)/kg

  1. How did you determine the P content in soils?

  1. The statistical comparison with the t-test is wrong. The t-test implies that the replicates of a treatment are independent from each other. For example, the litterfall in March is not a replicate of the April production. The Table 2 should be quoted without the letters of significance and without the standard errors.

  1. The seasonality test should be carried out with non parametric statistics because you have little data (use a Sign test instead of t-test and Spearman instead of Pearson).

  1. A Table containing the results (averages and variability) of soil analysis must be inserted.

  1. Clarify which data entered the PCA analysis

  1. Discussion. Delete the first paragraph. The statement is true for all forests not only for TDF.

Author Response

COMMENTS TO THE AUTHOR

AUTHOR´S RESPONSE IN MS. WITH TRACK CHANGES

Clarify what you mean by” Seasonally Dry Forests”?

Clarification is made to the reviewer. It is pointed out that the Tropical dry forests (TDFs) have been defined as a single biome occurring mostly in the lowlands where there is a marked period of drought during the year. Represents a terminology to describe these forest types (Pennintong et al. 2000).

Toby Pennington, R., Prado, D. E., & Pendry, C. A. (2000). Neotropical seasonally dry forests and Quaternary vegetation changes. Journal of Biogeography, 27(2), 261-273.

What do you mean by “Fine litterfall” Is there a such a thing as  “non Fine litterfall”?

It is explained to the reviewer that the term litterfall is used to indicate all materials that fall from the forest canopy and includes woody material up to 2 cm in diameter.

Non Fine litterfall is woody debris larger than 2 cm in diameter and is rarely studied because it contributes little to litterfall and nutrient release to the soil.

Materials and Methods. The mean temperatures and annual precipitation were derived from which period?

It is explained to the reviewer that the mean temperatures and annual precipitation were obtained from the long-term historical record (1958-2019). See Line:190-191.

Change the “Mountain landscape was characterized by the presence of high hills into “Mountain landscape is characterized by the presence of high hills into “

All changes were made as the reviewer suggested. L90-91

Change the “The Lomerio landscape was characterized by the presence of high hills into “The Lomerio landscape is characterized by the presence of high hills into..”

All changes were made as the reviewer suggested. L95-96

Change the “the soil properties..” into the soil properties of these sites are slightly acidic, have a sandy texture, have a low P content and a low cation….”

All changes were made as the reviewer suggested. This section was modified, in which a better description of the characteristics of the landscapes is made.

The concentrations of base cations in Table 1 are written as: cmol (+)/kg

All changes were made as the reviewer suggested. See Table 1.

How did you determine the P content in soils?

The changes proposed by the reviewer were accepted. The analytical method for the determination of phosphorus was included in the methodology section. L171-172

The statistical comparison with the t-test is wrong. The t-test implies that the replicates of a treatment are independent from each other. For example, the litterfall in March is not a replicate of the April production. The Table 2 should be quoted without the letters of significance and without the standard errors.

The reviewer's comments were not followed, since in this study the statistical tests consisted of comparing the annual means of the rates of each fine leaf litter fraction between landscapes, which do represent independent samples with replicates by plots. In the case of month-to-month inputs between fractions, these were compared graphically (Figure 3 and 4), and only with the objective of describing the seasonal patterns of each fraction.

The seasonality test should be carried out with non parametric statistics because you have little data (use a Sign test instead of t-test and Spearman instead of Pearson).

The reviewer's comments were partially accepted. Spearman's correlations were used instead of Pearson's correlation to look at the relationships between precipitation and fine litter fractions. On the other hand, it is noted that statistical tests (t-test, Sign test) were not used to evaluate the seasonality of the contributions of each fraction, but instead graphical comparisons were made to look only at the trend in these same patterns.

A Table containing the results (averages and variability) of soil analysis must be inserted.

All changes were made as the reviewer suggested. See Table 1.

Clarify which data entered the PCA analysis

We understand the reviewer's suggestion. We describe on lines 184-189 the variables used in the PCA analysis, including litterfall variables and structural and environmental characteristics.

Discussion. Delete the first paragraph. The statement is true for all forests not only for TDF.

We do not carry out the deletion suggested by the reviewer in this new version of the manuscript, since we consider that this paragraph is relevant to highlight the importance of studying the production of fine litter in tropical dry forests. However, we put it to the consideration of the editor of the Journal.

Round 2

Reviewer 1 Report

Argument about the scale for temperatures in Fig. 1 is not convincing.

Figure 5. Why is no phosphorous in the PCA?

LL328-329. Revise or reword this sentence, otherwise is just a speculation.

LL347-349. How come the contribution of leaf litter makes a species as a good candidate for restoration programs? Your data do not support this conclusion.

Author Response

COMMENTS TO THE AUTHOR

AUTHOR´S RESPONSE IN MS. WITH TRACK CHANGES

Reviewer # 1

Argument about the scale for temperatures in Fig. 1 is not convincing.

We understand the reviewer's suggestion. We clarify that for the construction of figure 1. We use technical diagrams commonly used in climate sciences, which are called Walter-Lieth climate diagrams, which use the principle of the Gaussen-Bagnouls aridity index. These climatological diagrams consist of representing the temperature and precipitation scales in a ratio of 1:2. In other words, the values of the temperature axis are half of the total values of the precipitation axis. For more details, see Maity, S. K. (2021), page 90-91.

Maity, S. K. (2021). Representation of Geographical Data Using Graphs. In Essential Graphical Techniques in Geography (pp. 47-152). Springer, Singapore.

We also used two articles as examples, which are presented below:

1. Castellanos-Barliza, J. C., Peláez, J. D. L., & Campo, J. (2018). Recovery of biogeochemical processes in restored tropical dry forest on a coal mine spoil in La Guajira, Colombia. Land Degradation & Development, 29(9), 3174-3183.

2. Seserman, D. M., Freese, D., Swieter, A., Langhof, M., & Veste, M. (2019). Trade-off between energy wood and grain production in temperate alley-cropping systems: an empirical and simulation-based derivation of land equivalent ratio. Agriculture, 9(7), 147.

Figure 5. Why is no phosphorous in the PCA?

We agree with the reviewer's concern. We already incorporated the phosphorus content in the soil in the PCA. As can be seen in the Figure, the vector that represents this variable overlaps with that of soil clay content and precipitation (ppt). However, the reviewer's comment is very pertinent, since this variable reflects part of the differences in soil fertility on which we develop the discussion of the differences between the two landscapes. L276-280.

LL328-329. Revise or reword this sentence, otherwise is just a speculation.

This sentence is not based on experimental evidence and does not contribute substantially to the development of the discussion and was therefore eliminated from the text. L330.

LL347-349. How come the contribution of leaf litter makes a species as a good candidate for restoration programs? Your data do not support this conclusion.

Several studies have suggested desirable attributes of plant species to be used in ecological restoration activities, some associated with their productivity. In particular, it has been pointed out that it is not only important that such species exhibit high growth rates, but also high litter production rates, and high litter decomposition rates. These two aspects, together, potentially favor the incorporation of greater amounts of organic matter and nutrients into the soil, contributing over time to the improvement of its physical-chemical and microbiological properties. Our study, in particular, focuses on the production rates of fine litter, which represents a functional parameter from which species can be comparatively analyzed, and support decision-making regarding the convenience of being selected for inclusion in dry forest ecosystem restoration programs. To support this statement, some studies are listed below:

Rai, A., Singh, A. K., Ghosal, N., & Singh, N. (2016). Understanding the effectiveness of litter from tropical dry forests for the restoration of degraded lands. Ecological Engineering, 93, 76-81.

Singh, K. P., Singh, P. K., & Tripathi, S. K. (1999). Litterfall, litter decomposition and nutrient release patterns in four native tree species raised on coal mine spoil at Singrauli, India. Biology and Fertility of Soils, 29(4), 371-378.

León, J. D., & Osorio, N. W. (2014). Role of litter turnover in soil quality in tropical degraded lands of Colombia. The Scientific World Journal2014.

In addition, the following sentence was included in the Discussion section:”Therefore, these attributes may project these species as potential for inclusion in future restoration programs”, to support what was exposed in Conclusions section. L 330-332; 348-352

Reviewer 2 Report

  1. Lines 173-174. The Bray method does not determine the total P but the available one. Correction must be made.

  1. Line 176. What is the Da? It seems that it is the bulk density. So the symbol Da is irrelevant.

  1. Lines 178-179. Change to : “The exchangeable Ca2+, Mg2+, K+ and Na+ were extracted with 1 M ammonium acetate solution at pH 7 and their concentrations were measured with flame atomic absorption spectroscopy.

  1. Table 1 should be moved to the Results section.

  1. Some comments should be made about the soil properties. For example from Table 1 it seems that soils are sandy for the Mountain landscape with a low organic matter and CEC.

Author Response

COMMENTS TO THE AUTHOR

AUTHOR´S RESPONSE IN MS. WITH TRACK CHANGES

Reviewer # 2

1. Lines 173-174. The Bray method does not determine the total P but the available one. Correction must be made.

The suggestions made by the reviewer were accepted. The respective corrections were made. L174

2. Line 176. What is the Da? It seems that it is the bulk density. So the symbol Da is irrelevant.

The reviewer's suggestion was accepted. L177 and Table 1

3. Lines 178-179. Change to: “The exchangeable Ca2+, Mg2+, K+ and Na+ were extracted with 1 M ammonium acetate solution at pH 7 and their concentrations were measured with flame atomic absorption spectroscopy.

The reviewer's suggestion were accepted and the changes were made. L179-181.

4. Table 1 should be moved to the Results section. consideramos

The reviewer's comments were not addressed. We clarify that in the previous revision, the reviewers asked us to describe in greater detail the soil characteristics that could reflect in greater detail the differences between the two types of landscapes studied. Therefore, in this new version we describe in greater detail these aspects that are not strictly experimental results. In fact, the more detailed description of the soils was intended to improve the understanding of the differences between the two landscapes, which could serve as a basis to analyze the possible differences found in the experimental results presented in the Results section. Therefore, we consider that incorporating this description of the study sites in the Results section is inconvenient and detracts from the coherence of the presentation of the work and specifically our results. We insist that the information in Table 1 is strictly a description of the study area and not of the results.

5. Some comments should be made about the soil properties. For example from Table 1 it seems that soils are sandy for the Mountain landscape with a low organic matter and CEC.  

We clarified that these comments had already been made on the following lines 105-111. However, considering the reviewer's comment we complemented the description for some parameters such as soil bulk density and exchangeable soil bases for the two types of landscape.
